# Molecular Mechanistic Pathways Targeted by Natural Compounds in the Prevention and Treatment of Diabetic Kidney Disease

**DOI:** 10.3390/molecules27196221

**Published:** 2022-09-21

**Authors:** Kaixuan Zhou, Xue Zi, Jiayu Song, Qiulu Zhao, Jia Liu, Huiwei Bao, Lijing Li

**Affiliations:** College of Pharmacy, Changchun University of Chinese Medicine, Changchun 130117, China

**Keywords:** diabetic kidney disease, natural compounds, oxidative stress, inflammation, renal fibrosis, mechanism

## Abstract

Diabetic kidney disease (DKD) is one of the most common complications of diabetes, and its prevalence is still growing rapidly. However, the efficient therapies for this kidney disease are still limited. The pathogenesis of DKD involves glucotoxicity, lipotoxicity, inflammation, oxidative stress, and renal fibrosis. Glucotoxicity and lipotoxicity can cause oxidative stress, which can lead to inflammation and aggravate renal fibrosis. In this review, we have focused on in vitro and in vivo experiments to investigate the mechanistic pathways by which natural compounds exert their effects against the progression of DKD. The accumulated and collected data revealed that some natural compounds could regulate inflammation, oxidative stress, renal fibrosis, and activate autophagy, thereby protecting the kidney. The main pathways targeted by these reviewed compounds include the Nrf2 signaling pathway, NF-κB signaling pathway, TGF-β signaling pathway, NLRP3 inflammasome, autophagy, glycolipid metabolism and ER stress. This review presented an updated overview of the potential benefits of these natural compounds for the prevention and treatment of DKD progression, aimed to provide new potential therapeutic lead compounds and references for the innovative drug development and clinical treatment of DKD.

## 1. Introduction

Diabetic kidney disease (DKD) is one of the most common complications of diabetes, and by 2040, more than 600 million people worldwide are expected to have diabetes, of whom 30–40% will develop DKD. It has become the major cause of chronic kidney disease in many developed and developing countries [1,2]. DKD is the most common cause of end-stage renal disease (ESRD) worldwide, chronic kidney disease is the dominant contributor to excess mortality in patients with type 2 diabetes [3]. Recently, sodium-glucose cotransporter-2 inhibitors (SGLT2i), glucagon-like peptide-1 receptor agonists (GLP-1 RAs), and dipeptidyl peptidase-4 inhibitors (DPP-4i) have been considered as new therapeutic options for DKD in the guidelines and consensus of some countries [4,5,6,7]. They have been proven to significantly improve the control of blood glucose and hemoglobin A1c (HbA1c), and prevent the onset of severe increased albuminuria [8]. But the use of these drugs has some limitations and adverse reactions. The adverse reactions of SGLT2i include genital mycotic infections and diabetic ketoacidosis [9,10]. The adverse reactions of GLP-1 RAs mainly include gastrointestinal damage and several cases of acute tubular injury [11]. DPP-4i causes sympathetic activation and enhanced Ca++/calmodulin-dependent protein kinase II signaling, which increases the risk of heart failure and arrhythmia in diabetic patients [12].

In recent years, some studies have shown that some natural compounds have the effects of lowering blood glucose, lowering blood lipid and anti-oxidation, and have good effects in the treatment of diabetes complications [13,14]. In this review, we searched the PubMed database and Google scholar with “(Diabetes nephropathy) OR (Diabetic kidney disease)”, found a recently published article on the treatment of DKD with a natural compound, and then searched with “((Diabetes nephropathy) OR (Diabetic kidney disease)) AND (compound)” to find all the studies on the treatment of DKD with this compound, and selected articles with in-depth mechanism research for review.

In this review, we aim to provide updated and comprehensive insights into the in vitro and in vivo experimental of natural compounds for DKD treatment and emphasize the potential mechanisms and molecular targets, especially those signaling pathways involved in metabolism regulation, anti-oxidation, anti-inflammatory, and anti-fibrosis, provide new potential therapeutic lead compounds and references for the innovative drug development and clinical treatment of DKD.

## 2. Signaling Pathways for DKD Progression

DKD is considered to be one of the serious long-term complications of diabetes affecting microvascular. Although DKD is a multifactorial disease with complex mechanisms, its pathogenesis could be initially explained by glucose and lipid metabolism disorders [15,16]. In diabetes, reactive oxygen species (ROS) is one of the important factors leading to renal injury. NADPH oxidase is widely expressed in the kidney and is the main source of oxidative stress in the kidney [17]. NADPH oxidase 4 (NOX4) is the main NADPH isoform in the kidney, which produces H_2_O_2_ that regulates physiological functions [18]. Hyperglycemia, hyperlipidemia, and other stimuli upregulate NOX4 expression in renal cells, leading to excessive ROS production [19,20]. Oxidative stress is related to the recruitment of inflammatory cells, inducing the production of proinflammatory factors, which leads to the activation of a variety of inflammatory pathways, such as nuclear factor-kappaB (NF-κB) signaling pathway, NACHT, LRR, and PYD domains-containing protein 3 (NLRP3) inflammasome signaling pathway, and other inflammatory pathways [21,22,23]. Proinflammatory factors can stimulate the expression of cytokines such as transforming growth factor-beta (TGF-β) from multiple pathways [24]. TGF-β is a key factor leading to fibrosis in most chronic kidney diseases, and the Smad signal is the key downstream regulator. Smad3 is strongly activated during fibrosis formation, while Smad7 is down-regulated, thereby promoting the expression of collagen I, collagen IV, and fibronectin (FN), resulting in increased extracellular matrix (ECM) production and decreased ECM degradation [25,26]. Under oxidative stress, the increase of ROS levels will lead to mitochondrial DNA breakage, causing a decrease in ATP production, which will affect the physiological function of the kidney [27]. In addition to oxidative stress-mediated kidney damage, endoplasmic reticulum (ER) stress also plays an important role in the development of DKD. Hyperglycemia and free fatty acids disturb the proteostasis, which leads to the accumulation of unfolded/misfolded proteins in the ER lumen, under ER stress, activating transcription factor 4 (ATF4), inositol requiring enzyme 1alpha (IRE1α) and protein kinase R-like endoplasmic reticulum kinase (PERK) are activated, which leads to the activation of downstream pro-apoptotic genes (caspase-4 or caspase-12), resulting in apoptosis [28]. Nuclear factor erythroid 2-related factor 2 (Nrf2) is a transcription factor that regulates genes that encode antioxidants, and it works in the nucleus. Kelch-like ECH-associated protein 1 (Keap1) binds to Nrf2 in the cytoplasm, preventing it from entering the nucleus. In the process of kidney diseases such as DKD, excessive ROS causes renal function damage, Keap1 increases, and Nrf2 activation decreases, reduce the production of antioxidant related proteins, such as superoxide dismutase(SOD)-1, heme oxygenase (HO)-1, and catalase (CAT) [29]. Autophagy can suppress inflammasome activity and clear damaged organelles in cells [30]. When kidney cells are exposed to oxidative stress and ER stress, autophagy activation plays a vital role in cell survival [31]. Autophagy also participates in the intracellular degradation of collagen, prevents the continuous accumulation of ECM, and plays a crucial role in inhibiting renal fibrosis [32]. Autophagy is mainly regulated by the mechanistic target of rapamycin (mTOR) signaling pathway, and its upstream is managed by AMP-activated protein kinase (AMPK). In DKD, renal AMPK activation is inhibited, which leads to the reduction of autophagy and the inability to clear damaged organelles [33].

The development of DKD can be roughly described as follows: in the early stage of DKD, hyperglycemia and the disorder of glucose and lipid metabolism caused by hyperglycemia gradually promote the structural and functional changes in the kidney, such as hyperfiltration, basement membrane thickening, glomerular mesangial matrix expansion, glomerular and tubular hypertrophy, and microalbuminuria [34,35,36]. Hyperglycemia is frequently accompanied by hyperlipidemia in the development of diabetes [37], hyperglycemia and hyperlipidemia can increase oxidative stress in the kidney, which is related to the reduction of antioxidant enzyme activity and the excessiveness of ROS [38,39]. ROS can directly impair mitochondrial function and can also regulate the expression of multiple genes associated with inflammation by activating NF-κB, ultimately leading to cellular inflammatory damage [40,41]. Oxidative stress and inflammation further lead to kidney damage, resulting in enhanced synthesis, weakened degradation, and excessive deposition of ECM, leading to renal fibrosis. Renal fibrosis is considered to be one of the most critical processes for DKD from a metabolic disorder, oxidative stress, and inflammation to ESRD [42,43] (Figure 1).

## 3. Natural Compounds

### 3.1. Phenolics

Phenolics are a large group of phytochemicals, with one or more aromatic rings and with one or more hydroxyl functional groups attached. They are present in fruits, cereals, vegetables, spices, teas, flowers and medical plants [44]. In the past decades, phenolics have been studied for their potential involvement in many areas including cancer, inflammation and microbial diseases [45].

Oleuropein is the most prevalent polyphenol in olive, which is a natural antioxidant molecule with a variety of biological activities and has many positive effects on human health, including anti-dyslipidemia, antidiabetic, anti-inflammatory, and antiatherogenic [46]. Studies have shown that oleuropein protected islet beta-cells from H_2_O_2_-induced cytotoxicity and promoted insulin secretion [47,48]. It has also been proven to treat gestational diabetes mellitus by activating AMPK signaling to improve lipid metabolism and inflammation [49]. In the treatment of DKD, Liu et al. found that administration of oleuropein can reduce renal injury, oxidative stress, and inflammation in db/db mice, oleuropein inhibits renal cell apoptosis by regulating the expression of mitogen-activated protein kinases (MAPK) signaling pathway and its downstream targets caspase-3, Bcl-2, and Bax [50]. In a clinical investigation, olive leaf extract (containing 136 mg oleuropein) effectively lowered plasma total cholesterol, low-density lipoprotein cholesterol, and triglyceride levels in prehypertensive patients. Furthermore, toxicology experiments revealed that olive leaf extract had no toxic effects at high doses [51].

Resveratrol is a natural polyphenol compound found in fruits such as grapes, mulberries, raspberries, and blueberries [52]. Many studies have indicated that resveratrol improves insulin sensitivity in insulin resistance animal models and has an anti-diabetic effect [53,54]. Clinical studies have shown that resveratrol can significantly reduce inflammation and oxidative stress, improve blood glucose control, and improve renal function in diabetes patients [55,56]. Resveratrol attenuates high-glucose (HG)-induced ER stress on the NRK 52E cells injured by hyperglycemia in vitro. The mechanism is related to inhibiting the increase of glucose-regulated protein 78 (GRP78) and C/EBP-homologous protein (CHOP) expression levels in cells, and alleviating ER stress-induced cell apoptosis [57]. Resveratrol can also improve renal injury by inhibiting podocyte apoptosis in DKD mediated by oxidative stress, the therapeutic effect is related to activating the AMPK signaling pathway [58]. Excessive generation of mitochondrial ROS is considered to be initiating event in the development of DKD. Resveratrol can activate sirtuin1 (SIRT1), increase the expression of SIRT1 and peroxisome proliferator-activated receptor gamma coactivator 1-alpha (PGC-1α) in renal tissue of DKD mice, reduce the production of mitochondrial ROS, and increase mitochondrial membrane potential, thus improving podocyte damage in diabetic mice [59]. Nrf2 is an important endogenous antioxidant transcription factor that protects cells from ROS injury, several studies have indicated that Nrf2 activators such as resveratrol have protective effects against DKD, resveratrol reduces oxidative stress, cell hyperproliferation, and ECM accumulation in mouse glomerular mesangium by activating the Keap1/Nrf2 signaling pathway [60]. In addition, Qiao et al. found that resveratrol inhibits HG-induced FN expression and reduce renal fibrosis by reducing MAPK activation and TGF-β1 expression in mesangial cells [61]. Gu et al. found that resveratrol can reduce the lipid accumulation in the kidney of DKD and improve kidney injury, the therapeutic effect of resveratrol is related to SIRT1 signaling pathway [62]. Another study found that resveratrol improves lipid metabolism in STZ-induced DKD by inducing AMPK/mTOR-mediated autophagy, at the same time, resveratrol alleviates lipid dysregulation by increasing the level of lipid oxidation-related protein such as peroxisome proliferator-activated receptor-alpha (PPARα) and carnitine palmitoyltransferase I (CPT1), and reducing the level of lipid production related protein such as sterol regulatory element-binding protein(SREBP)-1c and acyl-CoA synthetases (ACS) in DKD [63]. Some studies have demonstrated that resveratrol is a well-tolerated and safe compound in humans, but others have noted the harmful effects of resveratrol, which exhibited inhibition of P450 cytochromes when large doses were provided, while low doses of resveratrol are usually associated with beneficial effects, which needs to be considered in the research [64,65].

Gastrodin is a phenolic glycoside, which is the main active component of a traditional Chinese herbal medicine called *Gastrodia elata* [66]. Gastrodin has antioxidant, anti-inflammatory, hypolipidemic, anti-fatty liver, and therapeutic diabetes pharmacological activities. It has a therapeutic effect on the complications caused by diabetes, which can improve cognitive dysfunction, reduce blood glucose, reduce blood lipids, and increase insulin sensitivity in diabetic animals [67,68,69,70]. Huang et al. found that gastrodin increased the activity of antioxidant enzymes and reduced the level of inflammatory factors, thus reducing the inflammation, oxidative stress, and apoptosis of MPC-5 cells induced by HG. The mechanism is related to activating the AMPK/Nrf2 signaling pathway and reducing the formation of NLRP3 inflammasome [71]. Gastrodin is relatively safe to use. In mice, oral administration of gastrodin at a dose of 5000 mg/kg caused no mortality or obvious toxic effects [72]. At present, gastrodin is mainly used in clinics for neurasthenia, cephalagra, and other diseases, which has the effect of improving blood circulation [73]. With the deepening of research, gastrodin preparation is gradually applied to the treatment of diabetes, which improves the symptoms of patients [74]. Table 1 summarizes the available data on phenolics’ activities and the mechanisms of their action.

### 3.2. Alkaloids

Alkaloids are usually colorless and bitter basic nitrogen compounds (mainly heterocyclic), which mainly exist in the plant kingdom and usually have physiological activities [75]. Alkaloids exhibit different biological activities, such as anti-tumor, anti-inflammatory, antibacterial, and antiviral [76].

Trigonelline is an alkaloid in the extract of *Trigonella foenum-graecum*, *Coffea* sp., *Glycine max*, and *Lycopersicon esculentum*, which has a variety of biological activities such as the treatment of hyperglycemia, hypercholesterolemia, hormonal disorders, and cancers [77]. Studies showed that trigonelline significantly alleviated the oxidative stress and pathological changes in the kidneys and reduced the expression of FN and collagen ΙV in the mesangial ECM in DKD rats [78]. Human mesangial cells (HMCs) were stimulated with HG and treated with trigonelline. The results showed that trigonelline significantly inhibited the hyperproliferation of HMCs induced by HG and suppressed the levels of FN and collagen ΙV. Furthermore, trigonelline inhibited the activation of the Wnt/β-catenin signaling pathway to suppress cell-cycle progression and reduce apoptosis [79]. Li et al. found that trigonelline increased peroxisome proliferator-activated receptor-gamma (PPARγ) and glucose transporter type 4 (GLUT4) protein expression while suppressing leptin and tumour necrosis factor alpha (TNF-α) protein expression in the kidneys of DKD rats, thereby reduce inflammation, oxidative stress, and kidney cell apoptosis [80]. Chen et al. found that trigonelline upregulated the expression of miR-5189-5p, decreased hypoxia inducible factor 1 subunit alpha inhibitor (HIF1AN), and then activated the AMPK signaling pathway, increased autophagy level, and protected renal mesangial cells [81]. Toxicological studies showed that the lethal dose (LD_50_) of trigonelline in rats was around 5000 mg/kg after oral and subcutaneous administration [77]. In mice, trigonelline was fed 50 mg/kg daily for 21 days, and there was no change in the weight of the liver, kidney, thymus, thyroid, or adrenal gland [82]. Furthermore, trigonelline has good absorption and bioavailability. However, the existing data is insufficient to recommend trigonelline as a new medication; further clinical trials are needed to evaluate its adverse effects, pharmacokinetic characteristics, and mechanism of action.

Berberine is an isoquinoline alkaloid that occurs in *Coptis chinensis*. It has various pharmacological properties such as antioxidant, cardioprotective, anti-inflammatory, antibacterial, antidiabetic, and anticancer [83,84,85,86]. A meta-analysis of clinical studies shows that berberine significantly reduced fasting blood glucose, HbA1c, and triglyceride (TG), and improved insulin resistance in patients [87]. Studies have shown that berberine can significantly reduce fasting blood glucose, improve renal function, and alleviate podocyte injury in DKD rats. The mechanism may be related to berberine reducing the expression of phosphoinositide 3-kinase (PI3K) and protein kinase B (Akt) in the kidneys of DKD rats and increasing the expression of podocyte functional proteins (nephrin, podocin, and α3β1) in podocytes stimulated by high glucose [88]. In addition, berberine can also reduce the expression of p-AS160 and the level of membrane glucose transporter type 1 (GLUT1) by inhibiting the PI3K/Akt signaling pathway, thus reducing the glucose uptake of glomerular mesangial cells (GMCs) [89]. But another study found that in hypoxia/HG-induced NRK 52E and HK-2 cells, berberine promoted the activation of the PI3K/Akt signaling pathway and increased the expression of hypoxia-inducible factor-1alpha (HIF-1α), which led to a reduction in the apoptosis of cells [90].This may be related to the treatment of hypoxia. When glycolipid metabolism is out of balance, the lipid accumulation of renal tubular epithelial cells increases, leading to their dysfunction and tubulointerstitial fibrosis. Sun et al. study showed that berberine ameliorated apoptosis and decreased lipid accumulation in palmitate (PA)-induced HK-2 cells [91]. Rong et al. studied type 2 diabetic db/db mice and HG-induced HK-2 cells and found that berberine decreased the expression of alpha smooth muscle actin (α-SMA), collagen I, collagen IV, FN, and TGF-β1, thus reducing renal tubulointerstitial injury and renal fibrosis in diabetic db/db mice. Berberine increased the expression of CPT1, acyl-CoA oxidase 1 (ACOX1), and PPARα levels, thereby reducing lipid accumulation in the DKD models. The imbalance of glycolipid metabolism impairs mitochondrial morphology and mitochondrial function. PGC-1α is a transcriptional coactivator and has been shown to regulate mitochondrial functions, berberine enhanced AMPK activation and promoted PGC-1α expression in tubular epithelial cells [92]. In addition, berberine activated the CCAAT enhancer-binding protein beta (C/EBPβ) expression in HG-induced HK-2 cells. The C/EBPβ could combine with the reaction element on the promoter of lncRNA Gas5 to promote its expression, thereby inhibiting the miR-18a-5p expression, the expression level of miR-18a-5p is positively correlated with the ratio of apoptosis and mitochondrial ROS level. Meanwhile, C/EBPβ can activate the expression of PGC-1α and improve the mitochondrial energy metabolism. Berberine inhibited the generation of ROS, regulated the energy metabolism of mitochondria, and reduced apoptosis by activating the C/EBPβ/Gas5/miR-18a-5p signaling pathway and the C/EBPβ/PGC-1α signaling pathway [93]. Qin et al. found that in diabetic kidneys and PA-induced podocytes, berberine increased the protein levels of p-AMPK, PGC-1α, CPT1, and p-ACC while downregulating the expression of CD36. Therefore, it improves lipid accumulation, excessive generation of mitochondrial ROS, and mitochondrial dysfunction [94]. Another study showed that berberine also reduced the expression of dynamin-related protein 1 (Drp1) and consequently decreased the mitochondrial fission protein (MFF), mitochondrial fission protein 1 (Fis1), and mitochondrial dynamics proteins (Mid49, Mid51), thus to improving ROS generation, apoptosis, and mitochondrial dysfunction in diabetic kidneys and PA-induced podocytes [95]. Early studies have shown that Berberine inhibited the activation of the Notch/snail signaling pathway and upregulated α-SMA and E-cadherin levels in the DKD models, thus inhibiting renal tubular epithelial-mesenchymal transition (EMT) and renal fibrosis [96]. Berberine can also regulate the mTOR/P70S6K/4EBP1 signaling pathway and the toll-like receptor 4 (TLR4)/NF-κB signaling pathway in HG-induced podocytes, which activates autophagy and reduces inflammation and apoptosis [97,98]. Berberine has been demonstrated in animal experiments to have very minimal toxicity and adverse effects [99]. Some clinical trials on the safety of berberine in humans have found only mild gastrointestinal effects, such as diarrhea and constipation [100]. However, berberine has low intestinal absorption and oral bioavailability due to self-aggregation in the acidic environment of the stomach and intestinal first-pass elimination [101,102]. Therefore, it is necessary to change the dosage form to improve the bioavailability of berberine.

Sinomenine is a morphinane-type isoquinoline-derived alkaloid that is extracted from the roots and stems of *Sinomenium diversifolius* (Miq.) [103]. Sinomenine has an anti-diabetes effect on gestational diabetes rats and STZ-induced diabetes rats, reducing fasting blood glucose and inflammation levels in animals [104]. Studies found that sinomenine has a protective effect on human renal glomerular endothelial cells (HRGEs) induced by HG. When in a high glucose environment, intracellular ROS increases, which activates the Rho-associated protein kinase (ROCK) signaling pathway, destroys the expression of zonula occludens-1 (ZO-1)/occludin, and increases cell permeability. Sinomenine can reduce ROS production by activating the Nrf2 signaling pathway, thus protecting kidney cells [105]. Zhang et al. found that sinomenine can also activate the C/EBPα/claudin-5 pathway, alleviating HG-induced dysfunction of HRGEs and reducing inflammation, which has also been verified in DKD rats [106]. Zhu et al. found that sinomenine increased intracellular antioxidant enzymes, protected HK-2 cells from H_2_O_2_ damage, and reduced apoptosis. In DKD rats, sinomenine regulated the janus kinase (JAK)/signal transducer and activator of transcription (STAT) signaling pathway, thereby inhibiting the expression levels of fibrosis related proteins and inflammation related proteins [107]. In clinical studies, sinomenine has a potent anti-inflammatory effect. It has been formulated into tablets and injections for the treatment of knee osteoarthritis [108,109]. With the application of sinomenine, its adverse reactions, such as allergic reactions, nausea, and vomiting, have gradually attracted attention, which may be related to the bidirectional regulation of histamine release by sinomenine [110]. Huang et al. found that the safety of sinomenine was related to sexual distinction. The LD50 of male rats was 72.29 mg/kg, while that of female rats was 805.69 mg/kg [111]. Sinomenine may be used to treat kidney diseases in the future, but its safety is an issue that must be resolved. Table 2 summarizes the available data on alkaloids’ activities and the mechanisms of their action.

### 3.3. Flavonoids

Flavonoids are found almost everywhere in plants. They are rich in seeds, fruits, flowers, and medicinal plants [112]. They are low molecular weight compounds having a basic 15-carbon flavone skeleton, C6-C3-C6, with two benzene rings (A and B) linked by a three-carbon pyran ring (C) [113]. Flavonoids are categorized into six primary types based on their structure: anthocyanins, flavan-3-ols, flavones, flavanones, isoflavones, and flavonols [114].

Naringenin is a flavanone found mainly in citrus fruits, it has revealed promising pharmacological activities including cardiovascular diseases, anti-diabetic, antimicrobial, antiviral, anticancer, and anti-inflammatory [115,116]. On the NRK 52E cells injured by hyperglycemia in vitro and the DKD model in vivo, naringenin treatment markedly reduced the excessive production of intracellular ROS and downregulated the expression of endoplasmic reticulum (ER) stress marker proteins, including p-PERK, eukaryotic initiation factor 2 alpha (eIF2α), X-box-binding protein 1 (XBP1s), ATF4, and CHOP, anti-ER stress to reduce apoptosis of renal cells in diabetes [117]. Studies have also shown that, naringenin markedly reduced the proliferation and alleviated the morphological changes of NRK 52E cells induced by HG in a dose-dependent manner, naringenin ameliorates the renal damage of DKD mice, reduces glomeruli and renal tubular lesions through modulation of peroxisome proliferators-activated receptors (PPARs) with subsequent normalization of cytochrome P450 4A (CYP4A) expression and increasing 20-hydroxyeicosatetraenoic acid (20-HETE) [118]. In addition, Yan et al. found that MicroRNA let-7a was down expressed in both DKD rats and 293T mesangial cells under high glucose conditions, naringenin can inhibit TGF-β1/Smad signaling pathway by increasing the expression of MicroRNA let-7a, repressing glomerular mesangial cells proliferation and accumulation of ECM, thereby preventing renal fibrosis [119]. A clinical study showed a single dose of less than 900 mg of naringenin was safe and well tolerated in humans [120]. But another study found that naringenin has a dose-dependent inhibitory effect on the reproductive function of adult male mice and shows a pro-oxidative effect in testicular tissue [121]. This requires further clinical research to solve the safety and effectiveness of naringenin in humans.

Quercetin is present in numerous fruits and vegetables, recent studies have shown that quercetin has beneficial therapeutic effects in improving inflammation, blood lipids, and diabetes [122]. A meta-analysis of DKD animal experiments showed that after quercetin treatment, renal function index (such as urinary protein, uric acid, urinary albumin and serum creatinine levels) improved significantly [123]. The hyperglycemic environment in diabetes patients leads to the increase of advanced glycation end products (AGEs) production, AGEs can bind to the collagen that makes up the glomerular basement membrane, disrupting the glomerular barrier [124]. A study showed that quercetin could significantly reduce AGEs levels in renal tissue and serum levels of TNF-α and interleukin (IL)-6, increasing the level of SOD and glutathione peroxidase (GSH-Px) in serum [125]. Podocytes injury is one of the leading causes of proteinuria in patients with DKD, Liu et al. found that quercetin can prevent glomerular damage in diabetic mice and repress podocyte apoptosis by inhibiting the EGFR signaling pathway [126]. A clinical study showed that the expression of miR-485-5p in peripheral blood of patients with DKD decreased significantly, through cell experiment, it was found that quercetin inhibited the expression of YAP1 by regulating the increase of miR-485-5p, thereby inhibiting HG-induced HMCs hyperproliferation, inflammation, and oxidative stress [127]. In addition, Du et al. found that quercetin can also reduce the expression of yes-associated protein 1 (YAP1) by activating Hippo signaling pathway [128]. Dyslipidemia is one of the most serious and frequently occurring complications in DKD patients, lipid accumulation in the kidney has also been considered to play a role in the pathogenesis of DKD [129], Jiang et al. found that there were a large number of lipid droplets of different sizes in the renal cortex of diabetes mice, and quercetin could effectively reverse the lipid accumulation in both glomerulus and renal tubular cells by SREBP cleavage activating protein (SCAP)-SREBP2-low-density lipoprotein receptor (LDLr) signaling pathway [130]. Wang et al. found that quercetin can reduce renal lipid accumulation by regulating the expression of PPARα, CPT1, organic cation/carnitine transporter 2 (OCTN2), and acetyl-CoA carboxylase 2 (ACC2), it can also reduce renal inflammation by regulating the activation of renal NLRP3 inflammasome/caspase-1/IL-1β/IL-18 signaling pathway [131]. Currently, there have been some clinical experiments to investigate the effect of quercetin on diabetic patients. Oral quercetin (250 mg/day) for 8 weeks could significantly improve the antioxidant status of participants. Single oral administration of quercetin (400 mg) can effectively inhibit postprandial hyperglycemia after maltose loading in T2DM patients [132]. However, its oral bioavailability is minimal, limiting its therapeutic use. This is a major issue that must be addressed in future research.

Icariin is an active ingredient extracted from the traditional Chinese medicine *Epimedium*, some studies have shown that icariin can significantly inhibit cell apoptosis and oxidative stress [133]. In DKD rats and HG-induced MPC-5 cells, icariin could upregulate Sesn2 expression to induce mitophagy and activate the Keap1-Nrf2/HO-1 axis to inhibit NLRP3-related inflammation [134], icariin can also lighten renal inflammation by suppressing the TLR4/NF-κB signaling pathway, thus reducing renal fibrosis [135]. Icariin reduces the accumulation of collagen and FN in mesangial cells induced by high glucose by inhibiting the production of TGF-β1 and inhibiting Smad and extracellular signal-regulated kinase (ERK) signals in a G protein-coupled estrogen receptor (GPER)-dependent manner [136]. Jia et al. found that icariin can induce autophagy and reduce renal fibrosis in the DKD models, and its mechanism is related to the reduction of miR-192-5p, its overexpression inhibited glucagon-like peptide 1 receptor (GLP-1R), induced p-mTOR expression, and increasing the expression of collagen I, α-SMA, and FN. Icariin can alleviate DKD renal fibrosis by restoring autophagy through the miR-192-5p/GLP-1R pathway [137]. In addition, Zang et al. found that miR-122-5p also plays a role in the development of DKD, miR-122-5p inhibited forkhead box protein P2 (FOXP2) transcription, resulting in decreased cell viability, and inhibit E-cadherin expression, increasing α-SMA expression. Icariin can inhibit the expression of miR-122-5p and promote FOXP2 transcription, thus reducing the renal injury of DKD [138]. Icariin is a diglycoside, which indicates it is difficult to absorb. The bioavailability of icariin can be improved by pharmaceutical methods such as inclusion with β-cyclodextrin and propyleneglycol (PG)-liposomes [139]. At present, icariin lacks strong clinical evidence to prevent and treat DKD. In-depth mechanism research and evaluation of its safety are still needed.

Cardamonin is a flavonoid found in *Alpinia*, which can resist oxidative damage and apoptosis in vitro [140,141]. Cardamonin can reduce the blood glucose level of T2DM mice, improve hepatocyte lipid deposition, and inhibit sodium/glucose cotransporter 1 (SGLT1) [142,143]. Methylglyoxal (MGO) is capable of combining with proteins to form AGEs, which promote the production of ROS and induce cell apoptosis [144,145]. Gao et al. research showed that cardamonin reduced apoptosis, inflammation, oxidative stress, and renal fibrosis of MGO-treated NRK 52E cells and diabetic rats, its mechanism is related to regulating PI3K/AKT and JAK/STAT signaling pathway, reducing caspase-3, Bax, NF-κB, FN, α-SMA, and TGF-β1 protein expression, and increasing Bcl-2 and vimentin expression [146].

Morin is a flavonoid existing in the Moraceae family, which can improve oxidative stress, inflammation, and lipid metabolism [147,148]. Morin can reduce H_2_O_2_-induced Madin-Darby canine kidney cells oxidative stress and DNA oxidative damage [149]. Mathur et al. found that high glucose exposure in DKD models upregulated pleckstrin homology domain leucine-rich repeat protein phosphatases 1 (PHLPP1) and promoted the nuclear retention of forkhead box protein O1 (FoxO1) through double minute 2 protein (MDM2), thus leading to aberration in renal gluconeogenesis and activation of the apoptotic cascade. On the contrary, PHLPP1 gene silencing enhanced Nrf2 expression and weakened FoxO1-regulated apoptosis. Treatment with Morin can effectively down-regulate the expression of PHLPP1, relieve oxidative stress, and reduce renal cell apoptosis [150]. Another study showed morin inhibited HG-induced ECM expression, ROS generation, and NOX4 expression in glomerular mesangial cells. The mechanism is related to the inhibition of p38 MAPK and JNK signaling pathways [151]. Morin has good safety. No mortality or abnormal manifestations were found in rats given large doses of morin (about 300–2400 mg/kg) for 13 weeks [152]. The majority of the glycosylated morin administered orally was not absorbed in the small intestine and flowed into the colon, where it is then metabolized by colonic microorganisms to morin aglycones, which are readily absorbed [153,154].

Hesperetin is a flavanone that is present in the peels of several citrus fruits, and research found it has a variety of biological activities such as antioxidation, anti-inflammation, and improve glycolipid metabolism [155,156,157]. Early studies discovered that hesperetin had a hypoglycemic impact due to α-glucosidase inhibition [158,159]. Recent research has found that hesperetin has a therapeutic effect on DKD. It can increase antioxidant enzymes, such as thiobarbituric acid reactive substances (TBARS), GSH-Px, and CAT, reduce inflammatory cytokines (TNF-α, IL-6) expression, and inhibit TGF-β and glycogen synthase kinase-3beta (GSK-3β) expression, thereby reducing renal oxidative stress, inflammation, and fibrosis [160]. Chen et al. found that hesperetin can increase Glo-1 expression by activating the Nrf2 signaling pathway, thus accelerating AGEs clearance and decreasing inflammatory cytokines expression in the kidney. Furthermore, hesperetin reduces collagen ΙV and FN expression in the kidney and improves renal fibrosis [161]. Hesperetin also has good safety and has no mutagenic, toxic, or carcinogenic effects on pregnant mice [162]. Although hesperetin has obtained positive findings on diabetes treatment in animal studies, its functional mechanism in humans remains to be elucidated.

Fisetin is a natural dietary flavonoid that mainly exists in various fruits and vegetables, such as apples, grapes, cucumbers, and onions [163]. Through kinetic and molecular docking studies, it was found that fisetin had a potential inhibitory effect on α-glucosidase, which was verified by in vitro experiments [164,165]. Fisetin also has antioxidant, anti-inflammatory, and lipid metabolism-regulating activities [166,167], which can prevent the development of diabetes cardiomyopathy, diabetic neuropathy, and diabetes encephalopathy [168,169,170]. Research has found that fisetin reduced the EMT process, alleviated HG-induced podocyte injury and STZ-induced DKD, which is related to the restoration of cyclin-dependent kinase inhibitor 1B (CDKN1B)/P70S6K mediated autophagy and the inhibition of the NLRP3 inflammasome [171]. Obesity-induced hyperlipidemia is an important factor in DKD injury. Ge et al. studied high-fat diet (HFD) mice and PA-treated HK-2 cells and found that fisetin can regulate the insulin receptor signaling pathway to improve insulin sensitivity, inhibit the NF-κB signaling pathway, and receptor-interacting serine-threonine kinase 3 (RIP3)/NLRP3 signaling pathway to reduce inflammation [172]. Table 3 summarizes the available data on flavonoids’ activities and the mechanisms of their action.

### 3.4. Terpenoids

Terpenoids have very diverse physical and chemical properties as well as numerous biological activities, which are characterized by a carbon number multiple of five [173]. According to the amount of carbon, terpenoids can be divided into monoterpenes (C10), sesquiterpenes (C15), diterpenes (C20), sesterterpenes (C25), triterpenes (C30), tetraterpenes (C40), and higher homologs.

Sclareol is a natural diterpene that is an antifungal specialized metabolite produced by clary sage, *Salvia sclarea* [174]. Sclareol has been proven in studies to have anti-cancer, antioxidant, and anti-inflammatory effects [175,176,177], as well as the ability to enhance insulin sensitivity and glucose tolerance, hence improving metabolism in obese mice [175]. Han et al. found that sclareol treatment significantly alleviated renal dysfunction, fibrosis, and the levels of inflammatory cytokines in DKD mice, and sclareol treatment was dose-dependent. Sclareol inhibited the inflammatory reactions via the MAPK-mediated NF-κB pathway [178].

Ponicidin, a tetracyclic diterpenoid active ingredient extracted from the phytomedicine *Rabdosia rubescens*, has a positive effect on the treatment of a variety of cancers by inhibiting pro-inflammatory cytokine TNF-α induced angiogenesis and EMT [179,180,181]. An et al. found that ponicidin treatment can effectively decrease the levels of ROS and MDA in the serum of DKD rats, improve lipid metabolism in animals, reduce renal fibrosis and reduce the expression of inflammatory factors TNF-α, IL-1β, IL-6 and NF-κB [182].

Triptolide is a diterpenoid extracted from *Tripterygium wilfordii Hook. f.* which has a variety of biological activities such as antioxidation, immunomodulatory, and anticancer [183,184,185]. Some studies have shown that triptolide plays a significant role in the treatment of DKD, and a meta-analysis showed that triptolide significantly reduced albuminuria, blood urea nitrogen, serum creatinine, and urinary albumin/creatinine ratio in DKD animals [186,187]. T-helper (Th) cells is an immune cell populations, which can be divided into Th1/Th2 cells according to the cytokines. Th1 and Th2 cells cooperate to maintain the relative balance of immune response, and the activation of Th1 cells induces the secretion of proinflammatory cytokines, leading to the aggravation of DKD [188]. Guo et al. found that triptolide regulates the expression of pro-inflammatory (Interferon-γ, IL-12, and TNF-α) and anti-inflammatory (IL-4 and IL-10) cytokines and restores the balance of Th1/Th2 cells, thus reducing macrophage infiltration and the expression of inflammatory cytokines in the kidney [189]. Autophagy plays a positive role in the treatment of DKD, studies have shown that triptolide down-regulates the expression of miR-141-3p and miR-188-5p, causing the up-regulation of phosphatase and tensin homolog (PTEN) expression, affecting the expression of downstream pyruvate dehydrogenase kinase isoform 1 (PDK1), Akt, and mTOR, thus improving the level of autophagy, which in turn reduces the proliferation and fibrosis of mesangial cells [190,191,192]. Other studies have shown that triptolide can inhibit the Wnt3α/β-catenin signaling pathway to improve HG-induced EMT of podocytes [193]. Triptolide can also downregulate the protein expression levels of NLRP3 and apoptosis associated speck-like protein containing a CARD (ASC), inhibit the activation of the NLRP3 inflammasome, reducing the downstream protein expression such as caspase-1, IL-1β, and IL-18, thus preventing podocyte inflammatory injury [194]. Ren et al. found that triptolide can inhibit the TGF-β/Smad signaling pathway to downregulate p-Smad3, and inactivate kindlin-2, thus preventing podocyte EMT and protecting the kidney [195]. Clinically, triptolide is mainly used in inflammation and autoimmune diseases. It has high oral bioavailability but low safety. It has serious cytotoxicity to the heart, liver, kidney, and other organs [196]. Further research is needed to closely examine the relationship between the function and toxicology of triptolide to reduce its toxicity. Table 4 summarizes the available data on terpenoids’ activities and the mechanisms of their action.

### 3.5. Saponins

Saponins are high molecular weight amphiphilic compounds with a lipophilic moiety of triterpenoid or steroid aglycon and a hydrophilic moiety of sugars (usually glucose, arabinose, rhamnose, and xylose) [197]. Saponins usually have protective effects on the cardiovascular system and therapeutic effects on diabetes [198].

Dioscin is a natural steroidal saponin that is isolated from the Dioscoreaceae family [199], in 2015, DA-9801 containing dioscin completed the Phase II clinical trial for the treatment of diabetic neuropathy in the United States [200], some experimental studies showed that dioscin has therapeutic effect on some complications caused by diabetes mellitus, which can reduce the vascular damage in the retina of db/db mice and alleviate glycolipid metabolic disorder of T2DM [201,202]. In the DKD studies, dioscin significantly ameliorated renal damage via antagonizing the activation of the TLR4/NF-κB pathway and the production of inflammatory cytokines [203]. In addition, Zhong et al. found that dioscin reduced ROS levels, enhanced antioxidant enzyme (SOD, CAT) activities, and reduced inflammatory cytokine (IL-1β, IL-6, TNF-α, NF-κB) expressions. Dioscin could significantly inhibit the increase of p-PERK, IRE1, p-IRE1, ATF4, CHOP, and Caspase-12 expression levels in kidneys, and alleviate ER stress-induced cell apoptosis. Dioscin could also regulate the expression of the AMPK/mTOR pathway to promote autophagy in the DKD. Mitophagy and mitochondrial fission/fusion belong to the process of mitochondrial quality and quantity control, dioscin improves the expression of PTEN-induced putative kinase 1 (PINK1), Drp1, p-Drp1, and mitofusin 2(MFN2) to relieve the disorder of mitochondrial [204]. In the clinic, some drugs with dioscin as the main component are used to treat cardiovascular diseases [205]. However, diosgenin is a poorly soluble drug, and its oral absolute bioavailability is only 0.2% [206]. Some researchers have formulated dioscin as a new nano-drug delivery system, which can improve its oral bioavailability and drug loading [207]. Dioscin may have potential hepatotoxicity; when 300 mg/kg of dioscin was administered to rats for 90 days, levels of alanine aminotransferase increased significantly [208]. Clinical experimental research should be given special consideration.

Ginsenoside Rb1 is a tetracyclic triterpene saponin extracted from *Panax* L., which has a variety of biological activities such as antioxidation, anti-inflammation, anti-arrhythmia, anti-shock, and anti-diabetic [209,210]. Ginsenoside Rb1 regulates glucose and lipid metabolism by improving insulin and leptin sensitivity [211]. Studies have shown that ginsenoside Rb1 significantly alleviated the oxidative stress and pathological changes in the kidneys and reduced the expression of FN and collagen ΙV in the mesangial ECM in DKD rats. The mechanism is related to downregulating miR-3550 expression and inhibiting the Wnt/β-catenin signaling pathway [78]. He et al. study showed that ginsenoside Rb1 improves mitochondrial damage, oxidative stress, and apoptosis of renal podocytes in DKD models. High glucose can increase the expression of NOX4, a member of the NADPH oxidase family, and induce excessive ROS production. Aldose reductase (AR), an NADPH-dependent oxidoreductase, is a key rate-limiting enzyme in the polyol pathway of glucose metabolism. Ginsenoside Rb1 can bind to AR and inhibit AR activity, thereby reducing the expression of downstream NOX4, inhibiting the generation of ROS, and preventing the activation of caspase-9 [212]. The clinical experiment showed that healthy people orally had red ginseng extract containing 75 mg of ginsenoside Rb1 once or red ginseng extract containing 23 mg of ginsenoside Rb1 for two weeks without experiencing any abnormal effects [213,214]. At present, there are many studies showing the positive effect of ginsenoside Rb1 in the treatment of diabetes, which needs to be systematically studied in the clinic. 

Platycodin D is a deglycosylated triterpene saponin, which is found in *Platycodon grandiflorum*, a traditional Chinese medicinal herb with medicine and food homology [215]. Platycodin D has potent anti-inflammatory effects and anti-organ fibrosis effects, and it is also a new AMPK activator that reduces obesity in db/db mice through regulating adipogenesis and thermogenesis [216,217,218]. Studies have shown that platycodin D can inhibit the glomerular basement membrane thickening and fibrosis in DKD rats, which is related to the regulation of the PI3K/Akt signaling pathway to reduce oxidative stress and inflammation in the kidney [219]. Ferroptosis is involved in the regulation of cell death in a variety of diseases, including ischemia-reperfusion injury, cancer, and kidney disease. Excess iron in cells inhibits cell function by producing ROS, eventually leading to cell death [220]. Huang et al. found that HG increased oxidative stress and iron levels in HK-2 cells, thus causing ferroptosis. Platycodin D alleviated HG-induced ferroptosis in HK cells by upregulating glutathione peroxidase 4 (GPX4), ferritin heavy chain (FTH1), solute carrier family 7 member 11 (SLC7A11), and down-regulating the expression of acyl-coA synthetase long chain family member 4 (ACSL4) and transferrin receptor protein 1 (TFR1) [221]. Toxicological studies showed that platycodin D at a single oral dose of 2000 mg/kg body weight showed no toxic effects on mice [222]. An in vitro hemolysis assay showed that platycodin D had no obvious hemolytic effect on rabbit erythrocytes at concentrations ranging from 2.5~10 μM [223]. Platycodin D has poor bioavailability. After oral administration of 10 mg/kg platycodin D in rats, the bioavailability was only 0.079% [224]. This is the main problem restricting its application. Table 5 summarizes the available data on saponins’ activities and the mechanisms of their action.

### 3.6. Other Compounds

Caffeoylisocitric acid is a rare cinnamic acid derivative, it is a condensed ester of an isocitric acid and a caffeoylic acid, which is first found in *Amaranthus cruentus* [225]. Studies have shown that in the range of 0.01 to 200 µM, caffeoylisocitric acid has no significant cytotoxicity on normal HMCs, it can activates Nrf2 signaling pathway and inactivates MAPK signaling pathway to attenuate oxidative stress, inflammation and accumulation of ECM in mesangial cells under high glucose [226].

Crocin is a water-soluble carotenoid extracted from saffron (*Crocus sativus* L.), which has anti-inflammatory and antioxidant effects [227]. Clinical trials suggest that crocin may regulate the serum lipid profile in patients with metabolic disorders [228]. Many studies have shown that crocin has a therapeutic effect on DKD. Crocin can reduce oxidative stress and apoptosis to protect renal tubular epithelial cells from high glucose damage, which is related to the activation of the SIRT1/Nrf2 pathway [229]. In vivo experiments, crocin significantly reduced fasting blood glucose and blood lipid levels in db/db mice, and decreased the production of ROS and the expression of inflammatory factors in the kidney. The mechanism was related to the increase in the expression levels of Nrf2, SOD-1, HO-1, and CAT, and inhibition of the NF-κB signaling pathway [230]. Zhang et al. research showed that crocin reduced renal oxidative stress and inflammatory factors (TNF-α, IL-1β, and IL-18) by inhibiting the NLRP3 inflammasome, thereby reducing the expression of renal fibrosis proteins (TGF-β, collagen I, and collagen IV) and protecting the kidney [231]. A clinical study showed that after three months of crocin treatment, the fasting blood glucose and glycosylated hemoglobin of patients with diabetes were significantly reduced compared with the placebo group [232]. Hosseinzadeh et al. reported no animal mortality after single-dose administration of crocin (tolerated dose of 3 g/kg, i.v. or i.p.) in mice. Rats were administered crocin daily (15–180 mg/kg, i.p.) for 21 days, and no abnormal changes were observed [233]. These results suggest that crocin could be a promising natural product for the treatment of DKD.

Fraxin is the main active component of *Fraxinus rhynchophylla Hance* and belongs to the coumarin family, which has the functions of scavenging radicals and antioxidation [234]. The study found that fraxin inhibits the expression of inflammatory fibrosis factors by increasing the antioxidant enzymes in HG-induced primary glomerular mesangial cells, Cx43 interacted with AKT and consequently regulated the Nrf2 signaling pathway, fraxin could activate the Nrf2 pathway by regulating the interaction between connexin 43 (Cx43) and Akt to reduce intracellular oxidative stress and ROS generation. In addition, fraxin reduced the degree of renal fibrosis in db/db mice by inhibiting the protein expression of FN and ICAM-1 [235]. Table 6 summarizes the available data on other compounds’ activities and the mechanisms of their action.

## 4. Discussion

As the prevalence of DKD has increased over the decades, it has become one of the most common kidney diseases worldwide, to date, there are no sufficiently effective drug treatments to reverse the onset of DKD or prevent it from progressing to more severe stages of the disease. At present, the treatment of DKD mainly focuses on reducing blood glucose. However, the development of DKD is the result of a variety of injury factors. Lowering blood glucose can control the progression of the disease in the early stages, but in the middle and late stages, it is difficult to successfully prevent the development of the disease by only regulating blood glucose, and other targeted drugs are required to improve kidney damage. Indeed, many natural compounds have been extensively researched and have demonstrated good pharmacological action in the treatment of DKD.

The mechanisms of natural compounds reviewed in this article are mainly studied through the Nrf2 signaling pathway, NF-κB signaling pathway, TGF-β signaling pathway, NLRP3 inflammasome, autophagy, glycolipid metabolism and ER stress. Some compounds have multiple pathways and targets in the treatment of DKD. However, it is unclear if this effect just influences the expression of a specific protein before affecting the expression of other pathway proteins, or whether it regulates the expression of many pathway proteins and plays a therapeutic role together. This is also a problem in the current research of most natural compounds. It is necessary to deeply and systematically study the mechanism of natural compounds through transcriptomics, proteomics, pathway inhibitors, and other methods.

Here, we reviewed the selected natural compounds with beneficial effects in DKD reported by preclinical studies. They are all natural monomer compounds from plants. Some of these compounds have undergone relatively comprehensive mechanism studies and shown positive effects on DKD treatment, but there is no clinical evidence, such as oleopein, trigonelline, naringenin, icariin, hesperetin, ginsenoside Rb1, and platycodin D. Some natural compounds have comprehensive mechanism studies and are applied to other diseases clinically. Whether they can also be applied to the treatment of DKD needs further discussion, such as gastrodin, sinomenine, triptolide, and dioscin. Some natural compounds have been applied to the treatment of DKD in clinical preliminary studies, such as resveratrol, berberine, quercetin, and crocin. Some natural compounds only show a preliminary positive effect on the treatment of DKD, and their specific effects need to be further studied, such as cardamonin, morin, fisetin, sclareol, ponicidin, caffeoylisocitric acid, and fraxin. At the same time, it should be noted that some natural compounds have low bioavailability or poor safety, which are the reasons hindering clinical application and need to be solved by researchers.

This review provided an updated overview of the potential benefits of these natural compounds for the prevention and treatment of DKD progression, aiming to provide new potential therapeutic lead compounds and references for the innovative drug development and clinical treatment of DKD. Many natural compounds have shown potential effects on the treatment of DKD. At present, most studies focus on the antioxidant and anti-inflammatory capabilities of these compounds, which can reduce the deterioration of DKD. In the future, it may be possible to combine natural compounds with existing hypoglycemic drugs to exert the antioxidant and anti-inflammatory capabilities of natural compounds while reducing glucose, so as to achieve better protection of the kidney.

## 5. Methodology

In this review, we searched the PubMed database (https://pubmed.ncbi.nlm.nih.gov/) and Google scholar (https://scholar.google.com/). In the methodology, the name of the natural compound is replaced by “NC”. Search keywords include “(Diabetes nephropathy) OR (Diabetic kidney disease)”, “((Diabetes nephropathy) OR (Diabetic kidney disease)) AND (NC)”, “(Diabetes) AND (NC)”, “(Clinical) AND (NC)”, “(Toxicology) AND (NC)”, “(Pharmacokinetics) AND (NC)”.

## Figures and Tables

**Figure 1 molecules-27-06221-f001:**
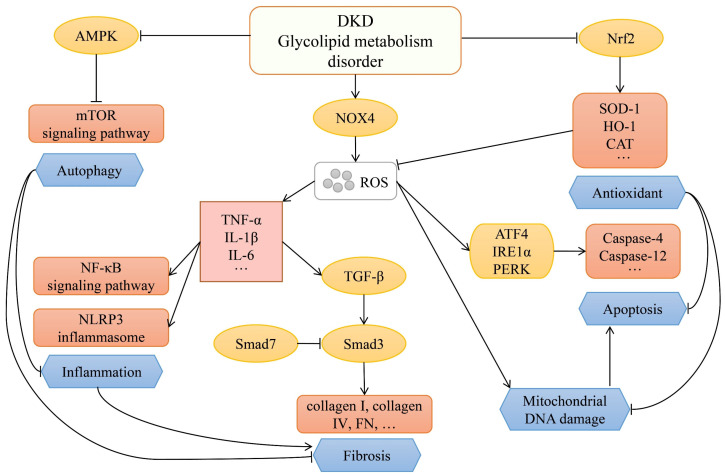
Some pathophysiological mechanisms of DKD.

**Table 1 molecules-27-06221-t001:** Mechanisms of phenolics in the treatment of DKD.

Natural Compound	Model	Function	Mechanism/Target	Reference
Oleuropein	In vivo: db/db mice DKD	Antioxidant Anti-inflammatory Reduce apoptosis	MAPK signaling pathway, caspase-3, Bcl-2, Bax	[50]
Resveratrol	In vitro: HG stimulated NRK 52E cells	Anti-ER stress Reduce apoptosis	GRP78, CHOP	[57]
	In vitro: HG stimulated mouse podocytes In vivo: db/db mice DKD	Antioxidant Reduce apoptosis	AMPK signaling pathway	[58]
	In vitro: HG stimulated mouse podocytes In vivo: db/db mice DKD	Antioxidant Reduce apoptosis	SIRT1, PGC-1α, NRF1, TFAM	[59]
	In vitro: HG stimulated SV40-MES-13 In vivo: STZ-induced mice DKD	Antioxidant Inhibit proliferation Anti-fibrotic	Keap1/Nrf2 signaling pathway	[60]
	In vitro: HG stimulated CRL-2573 In vivo: STZ-induced rats DKD	Anti-fibrotic	MAPK/TGF-β1 signaling pathway	[61]
	In vivo: STZ-induced mice DKD	Reduce lipid accumulation	SIRT1 signaling pathway	[62]
	In vivo: STZ-induced rats DKD	Reduce lipid accumulation Improve autophagy	AMPK/mTOR signaling pathway PPARα, CPT1, SREBP-1c, ACS	[63]
Gastrodin	In vitro: HG stimulated MPC-5 cells	Antioxidant Anti-inflammatory Reduce apoptosis	AMPK/Nrf2 signaling pathway NLRP3 inflammasome	[71]

**Table 2 molecules-27-06221-t002:** Mechanisms of alkaloids in the treatment of DKD.

Natural Compound	Model	Function	Mechanism/Target	Reference
Trigonelline	In vitro: HG stimulated HMCs In vivo: STZ-induced rats DKD	Antioxidant Reduce apoptosis Anti-fibrotic	Wnt/β-catenin signaling pathway FN, collagen IV	[78,79]
	In vivo: STZ-induced rats DKD	Antioxidant Anti-inflammatory Reduce apoptosis	PPARγ/GLUT4-leptin/TNF-α signaling pathway	[80]
	In vitro: HG stimulated HMCs	Improve autophagy	miR-5189-5p, HIF1AN AMPK signaling pathway	[81]
Berberine	In vitro: HG stimulated rat podocytes In vivo: STZ-induced rats DKD	Alleviate podocyte injury	PI3K/Akt signaling pathway nephrin, podocin, α3β1	[88]
	In vitro: HG stimulated GMCs In vivo: STZ-induced mice DKD	Reduce glucose uptake Inhibit proliferation	PI3K/Akt/AS160/GLUT1 signaling pathway	[89]
	In vitro: hypoxia/HG-stimulated NRK 52E and HK-2 cells	Reduce apoptosis	PI3K/Akt signaling pathway HIF-1α	[90]
	In vitro: PA stimulated HK-2 cells	Reduce apoptosis Reduce lipid accumulation	CPT1A, PPARα, PGC1α	[91]
	In vitro: HG stimulated HK-2 cells In vivo: db/db mice DKD	Anti-fibrotic Reduce lipid accumulation Improve mitochondrial function	α-SMA, collagen I, collagen IV, FN, TGF-β1, CPT1, ACOX1, PPARα, AMPK, PGC-1α	[92]
	In vitro: HG stimulated HK-2 cells In vivo: STZ-induced rats DKD	Antioxidant Reduce apoptosis Improve mitochondrial function	C/EBPβ/Gas5/miR-18a-5p signaling pathway C/EBPβ/PGC-1α signaling pathway	[93]
	In vitro: PA stimulated mouse podocytes In vivo: db/db mice DKD	Antioxidant Reduce lipid accumulation Improve mitochondrial function	AMPK/PGC-1α signaling pathway CPT1, ACC, CD36	[94]
	In vitro: PA stimulated mouse podocytes In vivo: db/db mice DKD	Antioxidant Reduce apoptosis Improve mitochondrial function	Drp1, MFF, Fis1, Mid49, Mid51	[95]
	In vitro: HG stimulated mouse podocytes In vivo: STZ-induced rats DKD	Anti-inflammatory Reduce apoptosis	TLR4/NF-κB signaling pathway	[97]
	In vitro: HG stimulated mRTEC In vivo: KKAy mice DKD	Anti-fibrotic	Notch/snail signaling pathway α-SMA, E-cadherin	[96]
	In vitro: HG stimulated mouse podocytes	Reduce apoptosis Improve autophagy	mTOR/P70S6K/4EBP1 signaling pathway	[98]
Sinomenine	In vitro: HG stimulated HRGEs	Antioxidant Decrease cell permeability	Nrf2 signaling pathway ROCK signaling pathway ZO-1, occludin	[105]
	In vitro: HG stimulated HRGEs In vivo: STZ-induced rats DKD	Anti-inflammatory Decrease cell permeability	C/EBPα/claudin-5 signaling pathway	[106]
	In vitro: H_2_O_2_ stimulated HK-2 cells In vivo: STZ-induced rats DKD	Antioxidant Reduce apoptosis Anti-inflammatory Anti-fibrotic	JAK/STAT signaling pathway	[107]

**Table 3 molecules-27-06221-t003:** Mechanisms of flavonoids in the treatment of DKD.

Natural Compound	Model	Function	Mechanism/Target	Reference
Naringenin	In vitro: HG stimulated NRK 52E cells In vivo: STZ-induced rats DKD	Antioxidant Anti-ER stress Reduce apoptosis	p-PERK, eIF2α, XBP1s, ATF4, CHOP	[117]
	In vitro: HG stimulated NRK 52E cells In vivo: STZ-induced mice DKD	Reduce renal tissue injury	PPARs, CYP4A, 20-HETE	[118]
	In vitro: HG stimulated 293T cells In vivo: STZ-induced rats DKD	Anti-fibrotic	MicroRNA let-7a TGF-β1/Smad signaling pathway	[119]
Quercetin	In vivo: STZ-induced rats DKD	Antioxidant Anti-inflammatory	AGEs, TNF-α, IL-6	[125]
	In vitro: HG stimulated mouse podocytes In vivo: db/db mice DKD	Reduce apoptosis	EGFR signaling pathway	[126]
	In vitro: HG stimulated HMCs In vivo: DKD patients	Inhibit proliferation Antioxidant Anti-inflammatory	miR-485-5p, YAP1	[127]
	In vitro: HG stimulated SV40-MES-13 In vivo: db/db mice DKD	Inhibit proliferation Antioxidant Anti-inflammatory	Hippo signaling pathway	[128]
	In vivo: db/db mice DKD	Reduce lipid accumulation	SCAP-SREBP2-LDLr signaling pathway	[130]
	In vivo: STZ-induced rats DKD	Reduce lipid accumulation Anti-inflammatory	PPARα, CPT1, OCTN2, ACC2 NLRP3 inflammasome/caspase-1/IL-1β/IL-18 signaling pathway	[131]
Icariin	In vitro: HG stimulated MPC-5 cells In vivo: STZ-induced rats DKD	Anti-inflammatory Improve autophagy Improve mitochondrial function	Sesn2, NLRP3 Nrf2 signaling pathway	[134]
	In vivo: STZ-induced mice DKD	Anti-inflammatory Anti-fibrotic	TLR4/NF-κB signaling pathway	[135]
	In vitro: HG stimulated SV40-MES-13	Anti-fibrotic	TGF-β1/Smad signaling pathway ERK, GPER	[136]
	In vitro: HG stimulated HK-2 and NRK 49F cells In vivo: STZ-induced rats DKD	Improve autophagy Anti-fibrotic	miR-192-5p/GLP-1R signaling pathway p-mTOR, collagen I, α-SMA, FN	[137]
	In vitro: HG stimulated NRK 52E cells In vivo: STZ-induced rats DKD	Reduce apoptosis Anti-fibrotic	miR-122-5p, FOXP2, E-cadherin, α-SMA	[138]
Cardamonin	In vitro: MGO stimulated NRK 52E cells In vivo: STZ-induced rats DKD	Antioxidant Anti-inflammatory Reduce apoptosis Anti-fibrotic	PI3K/AKT signaling pathway JAK/STAT signaling pathway caspase-3, Bcl-2, Bax, NF-κB, FN, α-SMA, TGF-β1, Vimentin	[146]
Morin	In vitro: HG stimulated NRK 52E cells In vivo: STZ-induced rats DKD	Antioxidant Reduce apoptosis	PHLPP1/FoxO1-Mdm2 signaling pathway Nrf2	[150]
	In vitro: HG stimulated primary rat GMCs	Antioxidant Anti-fibrotic	MAPK signaling pathway JNK signaling pathway NOX4	[151]
Hesperetin	In vivo: STZ-induced rats DKD	Antioxidant Anti-inflammatory Anti-fibrotic	TBARS, GSH-Px, CAT, TNF-α, IL-6, TGF-β, GSK-3β	[160]
	In vivo: STZ-induced rats DKD	Anti-inflammatory Anti-fibrotic	Nrf2 signaling pathway Glo-1, collagen ΙV, FN	[161]
Fisetin	In vitro: HG stimulated mouse podocytes In vivo: STZ-induced mice DKD	Anti-inflammatory Anti-fibrotic Improve autophagy	CDKN1B/P70S6K signaling pathway NLRP3 inflammasome	[171]
	In vitro: PA stimulated HK-2 cells In vivo: HFD-induced mice kidney injury	Improve insulin sensitivity Anti-inflammatory	Insulin receptor signaling pathway NF-κB signaling pathway RIP3/NLRP3 signaling pathway	[172]

**Table 4 molecules-27-06221-t004:** Mechanisms of terpenoids in the treatment of DKD.

Natural Compound	Model	Function	Mechanism/Target	Reference
Sclareol	In vitro: HG stimulated SV40-MES-13 In vivo: STZ-induced mice DKD	Antioxidant Anti-inflammatory Anti-fibrotic	MAPK/NF-κB signaling pathway	[178]
Ponicidin	In vivo: STZ-induced rats DKD	Antioxidant Anti-inflammatory Improve lipid metabolism Anti-fibrotic	TNF-α, IL-1β, IL-6, NF-κB	[182]
Triptolide	In vivo: STZ-induced rats DKD	Anti-inflammatory Regulate Th1/Th2 cells balance	Interferon-γ, IL-12, TNF-α, IL-4, IL-10	[189]
	In vitro: HG stimulated HMCs and HK-2 cells In vivo: STZ-induced rats DKD	Improve autophagy Inhibit proliferation Anti-fibrotic	miR-141-3p, miR-188-5p, PTEN, PDK1, Akt, mTOR	[190,191,192]
	In vitro: HG stimulated mouse podocytes	Anti-inflammatory Anti-fibrotic	Wnt3α/β-catenin signaling pathway NLRP3, ASC, caspase-1, IL-1β, IL-18	[193,194]
	In vitro: HG stimulated mouse podocytes In vivo: db/db mice DKD	Anti-fibrotic	TGF-β/Smad signaling pathway kindlin-2	[195]

**Table 5 molecules-27-06221-t005:** Mechanisms of saponins in the treatment of DKD.

Natural Compound	Model	Function	Mechanism/Target	Reference
Dioscin	In vivo: STZ-induced mice DKD	Anti-inflammatory	TLR4/NF-κB signaling pathway	[203]
	In vivo: STZ-induced rats DKD	Antioxidant Anti-inflammatory Anti-ER stress Reduce apoptosis Improve autophagy Improve mitochondrial function	IL-1β, IL-6, TNF-α, NF-κB, p-PERK, IRE1, p-IRE1, ATF4, CHOP, Caspase-12, PINK1, Drp1, p-Drp1, MFN2 AMPK/mTOR signaling pathway	[204]
Ginsenoside Rb1	In vivo: STZ-induced rats DKD	Antioxidant Anti-fibrotic	Wnt/β-catenin signaling pathway miR-3550	[78]
	In vitro: HG stimulated mouse podocytes In vivo: STZ-induced mice DKD	Antioxidant Protect mitochondria Reduce apoptosis	AR, NOX4, caspase-9	[212]
Platycodin D	In vivo: STZ-induced rats DKD	Antioxidant Anti-inflammatory Anti-fibrotic	PI3K/Akt signaling pathway	[219]
	In vitro: HG stimulated HMCs and HK-2 cells	Antioxidant Reduce ferroptosis	GPX4, FTH1, SLC7A11, ACSL4, TFR1	[221]

**Table 6 molecules-27-06221-t006:** Mechanisms of other compounds in the treatment of DKD.

Natural Compound	Model	Function	Mechanism/Target	Reference
Caffeoylisocitric acid	In vitro: HG stimulated HMCs	Antioxidant Anti-inflammatory Anti-fibrotic	Nrf2 signaling pathway MAPK signaling pathway	[226]
Crocin	In vitro: HG stimulated HK-2 cells	Antioxidant Reduce apoptosis	SIRT1/Nrf2 signaling pathway	[229]
	In vivo: db/db mice DKD	Antioxidant Anti-inflammatory	Nrf2, SOD-1, HO-1, CAT NF-κB signaling pathway	[230]
	In vivo: STZ-induced rats DKD	Antioxidant Anti-inflammatory Anti-fibrotic	NLRP3 inflammasome TNF-α, IL-1β, IL-18, TGF-β, collagen I, collagen IV	[231]
Fraxin	In vitro: HG stimulated primary GMCs In vivo: db/db mice DKD	Antioxidant Anti-inflammatory Anti-fibrotic	Nrf2 signaling pathway Cx43, Akt, FN, ICAM-1	[235]

## Data Availability

Not applicable.

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
