# Peer review of "Molecular Mechanistic Pathways Targeted by Natural Compounds in the Prevention and Treatment of Diabetic Kidney Disease"

_molecules, 2022, doi:10.3390/molecules27196221_

Round 1
Reviewer 1 Report
In my opinion, this review article is interesting and well written that aims to explore the potential benefits of natural products for preventing and treating DKD progression. The findings from this study may provide new potential therapeutic lead compounds and references for the innovative drug development and clinical treatment of DKD. However, I have some suggestions for the authors to consider in revising the manuscript:
1. You have mentioned that current therapy options for DKD have limitations. Please specify.
2. Please explain the phytochemicals of natural products, including phenolics, alkaloids, flavonoids, etc.
3. It would be better if the authors could provide a few more figures to summarise the mechanism of the natural product phytochemicals in the treatments of DKD.
4. Please improve the table arrangement.
Author Response
Dear Reviewer:
Thanks for your comments concerning our manuscript. Those comments are all valuable and very helpful for revising and improving our paper. We have studied the comments carefully and have made corrections which we hope meet with approval.
Response to comments:
- You have mentioned that current therapy options for DKD have limitations. Please specify.
Reply: The manuscript is supplemented with the possible adverse reactions of current DKD treatment drugs.
- Please explain the phytochemicals of natural products, including phenolics, alkaloids, flavonoids, etc.
Reply: Phytochemicals have been added at the beginning of each section of the manuscript.
- It would be better if the authors could provide a few more figures to summarise the mechanism of the natural product phytochemicals in the treatments of DKD.
Reply: Thanks very much for your advice. Figures can better show the therapeutic mechanism of compounds on diseases, but we hope to show the specific information (research methods and depth) of the cited article through tables so that readers can directly obtain the current research status of compounds in DKD treatment. Therefore, more tables are used in the manuscript.
- Please improve the table arrangement.
Reply: Some adjustments have been made, but due to the large amount of content, the aesthetics are not very good. I hope you can forgive me.
Thank you very much for reviewing this manuscript.
Reviewer 2 Report
The manuscript entitled " Molecular Mechanistic Pathways Targeted by Natural Products in the Prevention and Treatment of Diabetic Kidney Disease” is well written. The authors have reviewed the potential benefits of natural products for the prevention and treatment diabetic kidney disease. This review paper is well organized and interesting.
However, here are my comments and suggestion for further improvement of the manuscript. After all corrections, the manuscript is suitable for publication in molecules.
Abstract
1. Line 5: in vitro and in vivo must be in italic font.
Introduction
2. Second paragraph Line 5: italicize in vitro and in vivo. The authors should italicized those two scientific names throughout the manuscript.
3. In the introduction section, the criteria adopted in choosing the papers should be stated.
Methodology
6 A separate section namely Methodology' must be added just after section 4, depending on the journal format. It must contain all information related to article search and databases used to search research papers in the current review for a better understanding for the readers.
Conclusion
4. The manuscript should end with critical conclusions about the prospects, speculations and the real state regarding the applicability of the Natural Products in the prevention and treatment of diabetic kidney disease.
Sample Availability
5. This section is not important as the authors didn’t perform experiments or present research findings.
Thanks
Author Response
Dear Reviewer:
Thanks for your comments concerning our manuscript. Those comments are all valuable and very helpful for revising and improving our paper. We have studied the comments carefully and have made corrections which we hope meet with approval.
Response to comments:
Abstract
- Line 5: in vitro and in vivo must be in italic font.
Reply: These words in the manuscript have been modified.
- Second paragraph Line 5: italicize in vitro and in vivo.The authors should italicized those two scientific names throughout the manuscript.
Reply: These words in the manuscript have been modified.
- In the introduction section, the criteria adopted in choosing the papers should be stated.
Reply: More detailed article retrieval and selection scheme has been supplemented in the manuscript
Methodology
- A separate section namely Methodology' must be added just after section 4, depending on the journal format. It must contain all information related to article search and databases used to search research papers in the current review for a better understanding for the readers.
Reply: The methodology has been added after section 4 of the manuscript.
Conclusion
- The manuscript should end with critical conclusions about the prospects, speculations and the real state regarding the applicability of the Natural Products in the prevention and treatment of diabetic kidney disease.
Reply: We have revised the discussion and added prospects.
Sample Availability
- This section is not important as the authors didn’t perform experiments or present research findings.
Reply: Thank you very much for reviewing this manuscript.
Reviewer 3 Report
Reviewer’s recommendation
The manuscript was well written and discussed about the targets of natural compounds at molecular level in the prevention and treatment of diabetic kydney disease. I recommend this manuscript for publication on Molecules journal after the authors address all comments below
1. The authors should clarify “these compounds” in the abstract as in the previous sentences the authors did not mention any compounds.
2. The first sentence in Introduction is too long, it should be splitted into at least two sentences.
3. The sentence in Introduction “Recently, some countries’ DKD treatment guidelines and consensus have been modified, sodium-glucose cotransporter-2 inhibitors (SGLT2i), glucagon-like peptide-1 receptor agonists (GLP-1 RAs), and dipeptidyl peptidase-4 inhibitors (DPP-4i) show certain cardioprotective and renal protective effects when reducing blood glucose, which are considered to be new therapeutic options for DKD[4-7].” was too long and not clear, please rewrite this sentence.
4. What are the effects of sodium-glucose cotransporter-2 inhibitors (SGLT2i), glucagon-like peptide-1 receptor agonists (GLP-1 RAs), and dipeptidyl peptidase-4 inhibitors (DPP-4i) on DKD treatment ?
5. Subtitle at page 3 “Natural products” should be changed to “Natural compounds” that are more relavant.
6. Term “diabetes animals” in page 4 should be changed to “diabetic animals”.
7. “Natural product” in Table 1, 2, 3, 4, 5, 6 should be changed to “Natural compound”.
8. The sentence “Participants are given 500 mg of quercetin every day, which significantly lowers serum high-density lipoprotein cholesterol (HDL-C)...” in page 9 indicated that quercetin is not good for participants, as HDL-C is considered to be “good cholesterol”, what is the remark of authors about this statement ?
9. The sentence “Dioscin has low toxicity; when 300 mg/kg of dioscin was administered to rats for 90 days, levels of alanine aminotransferase increased significantly” in page 14 – 15 indicated that dioscin did not have low toxicity, as administered at quite low concentration 300 mg/kg level of alanine aminotransferase significantly increased, the author should re-write this statement.
1. The authors used term “PD” in page 15 without explanation, the authors should clarify this term.
1. The sentence “The concentration range is 2.5~10 μM platycodin D had no obvious hemolytic effect on rabbit erythrocytes” in page 15 showed the concentration of platycodin D in blood ?
1. The sentence “Although platycodin D has high safety, its bioavailability is poor” in page 15: based on which evidences the authors can conclude platycodin D has high safety. In this manuscript, the authors only showed no toxicity observed at a single oral dose of 2000 mg/kg body weight on mice. This evidence is not enough to make such conclusion.
1. Term “natural products” in Discussion should be changed into natural compounds. They are not products, they are chemical compounds derived from plants.

Author Response
Dear Reviewer:
Thanks for your comments concerning our manuscript. Those comments are all valuable and very helpful for revising and improving our paper. We have studied the comments carefully and have made corrections which we hope meet with approval.
Response to comments:
- The authors should clarify “these compounds” in the abstract as in the previous sentences the authors did not mention any compounds.
Reply: We changed this sentence to "these reviewed compounds," indicating that they are compounds reviewed in the manuscript.
- The first sentence in Introduction is too long, it should be splitted into at least two sentences.
Reply: This sentence has been modified.
- The sentence in Introduction “Recently, some countries’ DKD treatment guidelines and consensus have been modified, sodium-glucose cotransporter-2 inhibitors (SGLT2i), glucagon-like peptide-1 receptor agonists (GLP-1 RAs), and dipeptidyl peptidase-4 inhibitors (DPP-4i) show certain cardioprotective and renal protective effects when reducing blood glucose, which are considered to be new therapeutic options for DKD[4-7].” was too long and not clear, please rewrite this sentence.
Reply: This sentence has been modified.
- What are the effects of sodium-glucose cotransporter-2 inhibitors (SGLT2i), glucagon-like peptide-1 receptor agonists (GLP-1 RAs), and dipeptidyl peptidase-4 inhibitors (DPP-4i) on DKD treatment ?
Reply: The effect of these drugs on DKD treatment has been supplemented in the manuscript.
- Subtitle at page 3 “Natural products” should be changed to “Natural compounds” that are more relavant.
Reply: It has been changed. At the same time, the "natural product" in the article title and other parts has been changed to "natural compound" to make the full text consistent.
- Term “diabetes animals” in page 4 should be changed to “diabetic animals”.
Reply: This has been modified.
- “Natural product” in Table 1, 2, 3, 4, 5, 6 should be changed to “Natural compound”.
Reply: These have been modified.
- The sentence “Participants are given 500 mg of quercetin every day, which significantly lowers serum high-density lipoprotein cholesterol (HDL-C)...” in page 9 indicated that quercetin is not good for participants, as HDL-C is considered to be “good cholesterol”, what is the remark of authors about this statement ?
Reply: The cited article shows that HDL-C in both quercetin group and placebo group decreased, which may be caused by some other factors. For this imprecise statement, we have made changes in the manuscript and selected some other clinical trials for description.
- The sentence “Dioscin has low toxicity; when 300 mg/kg of dioscin was administered to rats for 90 days, levels of alanine aminotransferase increased significantly” in page 14 – 15 indicated that dioscin did not have low toxicity, as administered at quite low concentration 300 mg/kg level of alanine aminotransferase significantly increased, the author should re-write this statement.
Reply: We rewrite this statement, changing “Dioscin has low toxicity” to “Dioscin may have potential hepatotoxicity”
- The authors used term “PD” in page 15 without explanation, the authors should clarify this term.
Reply: I used abbreviation in the early stages of writing the manuscript and forgot to modify it. It has now been updated.
- The sentence “The concentration range is 2.5~10 μM platycodin D had no obvious hemolytic effect on rabbit erythrocytes” in page 15 showed the concentration of platycodin D in blood ?
Reply: This is an in vitro hemolysis analysis, which has been supplemented in the manuscript.
- The sentence “Although platycodin D has high safety, its bioavailability is poor” in page 15: based on which evidences the authors can conclude platycodin D has high safety. In this manuscript, the authors only showed no toxicity observed at a single oral dose of 2000 mg/kg body weight on mice. This evidence is not enough to make such conclusion.
Reply: The expression here is too absolute, so "platycodin D has high safety" is deleted in the manuscript.
- Term “natural products” in Discussion should be changed into natural compounds. They are not products, they are chemical compounds derived from plants.
Reply: These have been modified.
Thank you very much for reviewing this manuscript.